# Effectiveness of Brodalumab for the Treatment of Moderate-to-Severe Psoriasis: A Retrospective, Real-World Multicenter Study with a Focus on Obese and Multi-Failure Patients—IL PSO (Italian Landscape Psoriasis)

**DOI:** 10.3390/jcm14041087

**Published:** 2025-02-08

**Authors:** Diego Orsini, Dario Graceffa, Martina Burlando, Anna Campanati, Elena Campione, Claudio Guarneri, Alessandra Narcisi, Paolo Pella, Paolo Romita, Massimo Travaglini, Leonardo Zichichi, Luisa Maria Halina Arancio, Ginevra Baggini, Riccardo Balestri, Tommaso Bianchelli, Luca Bianchi, Alexandra Maria Giovanna Brunasso, Anna Elisabetta Cagni, Giacomo Caldarola, Gianluca Calianno, Anton Carpentieri, Martino Carriero, Andrea Carugno, Franco Cona, Antonio Costanzo, Emanuele Claudio Cozzani, Giacomo Dal Bello, Giovanni Carlo Lazzaro Danzuso, Annunziata Dattola, Marianna Donnarumma, Elena De Col, Maria Esposito, Carmen Silvia Fiorella, Marco Galluzzo, Francesca Graziola, Gaetano Licata, Matteo Licciardello, Agostina Legori, Piergiorgio Malagoli, Federica Mola, Gaia Moretta, Andrea Muracchioli, Attilia Musumeci, Maria Letizia Musumeci, Gianluca Pagnanelli, Vincenzo Panasiti, Lucia Peterle, Eugenio Provenzano, Marco Rubatto, Oriele Sarno, Davide Strippoli, Fabrizio Vaira, Maria Concetta Fargnoli

**Affiliations:** 1San Gallicano Dermatological Institute (IRCCS), 00144 Rome, Italy; diego.orsini@ifo.it (D.O.); mariaconcetta.fargnoli@ifo.it (M.C.F.); 2Section of Dermatology, Department of Health Sciences (DiSSal), University of Genoa, 16126 Genoa, Italyemanuele.cozzani@unige.it (E.C.C.); 3IRCCS San Martino University Hospital, 16132 Genoa, Italy; 4Dermatological Clinic, Department of Clinical and Molecular Sciences, Polytechnic Marche University, 60100 Ancona, Italytommaso.bianchelli@gmail.com (T.B.); 5Dermatology, Department of Systems Medicine, University of Rome Tor Vergata, 00133 Rome, Italyluca.bianchi@uniroma2.it (L.B.); marco.galluzzo@uniroma2.it (M.G.); 6Department of Biomedical and Dental Sciences and Morpho Functional Imaging, Section of Dermatology, University of Messina, 98122 Messina, Italy; 7Dermatology Unit, IRCCS Humanitas Research Hospital, 10134 Rozzano, Italyantonio.costanzo@hunimed.eu (A.C.); 8Dermatology Unit, Ospedale degli Infermi, 13875 Biella, Italy; 9Section of Dermatology and Venereology, Department of Precision and Regenerative Medicine and Ionian Area (DiMePre-J), University of Bari “Aldo Moro”, 70124 Bari, Italy; 10U.O.S.D. Dermatologica—Centro Per La Cura Della Psoriasi, Ospedale A. Perrino, 72100 Brindisi, Italy; 11Dermatology Unit, Ospedale S A Antonio Abate, ASP Trapani, 91016 Erice, Italy; l.zichichi@gmail.com (L.Z.);; 12Fondazione IRCCS Cà Granda Ospedale Maggiore Policlinico Milano, ASST Crema, 20157 Milan, Italy; 13Outpatient Clinic, Section of Dermatology, ASL AL, 15100 Alessandria, Italy; 14Division of Dermatology, Psoriasis Outpatient Service, APSS, 38122 Trento, Italy; riccardo.balestri@gmail.com; 15Department of Internal Medicine-Dermatology, Villa Scassi Hospital, ASL3, 16151 Genoa, Italy; 16Unità Operativa Dipartimentale di Dermatologia e Venereologia, IRCCS San Gerardo, 20156 Milan, Italy; 17Dermatology Unit, Agostino Gemelli University Polyclinic Foundation, IRCCS, 00168 Rome, Italy; giacomo.caldarola@unicatt.it; 18Section of Dermatology, Department of Translational Medicine and Surgery, Catholic University of the Sacred Heart, 00168 Rome, Italy; 19Dermatology Outpatient, ASL Taranto, 74100 Taranto, Italy; 20Dermatology Unit, Department of Medicine and Surgery, University of Insubria, 21100 Varese, Italy; andrea.carugno@uninsubria.it; 21Umberto I, City Hospital of Enna, 94100 Enna, Italy; 22Section of Dermatology, Department of Medicine, ASST di Mantova, 46100 Mantova, Italy; 23U.O.S.D. Dermatologia, A.O. ’S.Elia’—Caltanissetta, 93100 Caltanissetta, Italy; 24Dermatology Unit, Department of Clinical Internal, Anesthesiological and Cardiovascular Science, University of La Sapienza, 00161 Rome, Italy; 25Dermatology Unit, “Andrea Tortora” Hospital, ASL Salerno, 84098 Pagani, Italy; 26Unit of Dermatology, Civil Hospital of Imperia, 18039 Imperia, Italy; 27Department of Biotechnological and Applied Clinical Sciences, University of L’Aquila, 67100 L’Aquila, Italy; maria.esposito3@univaq.it; 28Division of Dermatology, Mons. Dimiccoli Hospital, 76121 Barletta, Italy; carmen.fiorella10@gmail.com; 29Dermatologic Clinic, AOU Maggiore della Carità Hospital, 28100 Novara, Italy; 30Section of Dermatology, Koelliker Hospital, 10126 Turin, Italy; matteo.licciardello@gmail.com; 31UO Dermatologia IRCCS Ospedale Galeazzi & Università degli Studi di Milano, 20157 Milan, Italy; 32Department of Dermatology, Dermatology Unit Azienda Ospedaliera San Donato Milanese, 20097 Milan, Italy; 33Dermatology Unit, ASST Lariana, 22020 Como, Italy; 34Department of Dermatology, Istituto Dermopatico dell’Immacolata IRCCS Roma, 00167 Rome, Italy; gaia.mor@hotmail.it (G.M.);; 35Unit of Dermatology, San Bartolomeo Hospital, 19042 La Spezia, Italy; 36U.O.S.D. Dermatologia, ARNAS Garibaldi, 95123 Catania, Italy; 37UOC Dermatologia, University of Catania, PO “G. Rodolico”, AOU Policlinico “G. Rodolico-San Marco”, 95123 Catania, Italy; 38Fondazione Policlinico Universitario Campus Bio Medico, 00128 Rome, Italy; 39Dermatology Unit, Azienda Ospedaliera di Cosenza, 87100 Cosenza, Italy; 40Department of Medical Sciences, Section of Dermatology, University of Turin, 10126 Turin, Italy; 41Department of Dermatology and Dermosurgery, AOSG San Giuseppe Moscati, 83100 Avellino, Italy; 42Dermatology Unit, Manzoni Hospital, ASST-Lecco, 23900 Lecco, Italy; d.strippoli@asst-lecco.it; 43Unit of Dermatology, IRCCS Ospedale San Raffaele, 20132 Milan, Italy

**Keywords:** brodalumab, psoriasis, obesity, interleukin-17 inhibitors, failure

## Abstract

**Background/Objectives**: Brodalumab is a monoclonal antibody against the anti-IL-17 receptor A, approved for patients with moderate-to-severe psoriasis. This retrospective study investigated patients in clinical practice to assess the impact of body weight and previous treatments with biologics on the effectiveness of brodalumab. **Methods**: Patients were treated according to clinical practice, and assessed at baseline, 16, 36 and 52 weeks by means of the Psoriasis Area Severity Index (PASI) and DLQI score. Overall, 299 patients were included (147 naïve to biologics). **Results**: Mean PASI was significantly reduced compared with the baseline in the overall population by week 4 and continued to decrease at each study time point (15.9 ± 7.9 at baseline, 5.4 ± 5.3 at week 4, 1.9 ± 3.6 at week 6, 1.0 ± 2.1 at week 36, and 0.8 ± 2.1 at week 52; *p* < 0.001 at each control). PASI improved significantly both in bio-naïve and bio-experienced patients (*p* < 0.001). The proportions of patients achieving PASI 75, PASI 90, and PASI 100 were comparable between the bio-naïve and bio-experienced groups at all time points. The percentages of patients who achieved PASI 75 were similar in obese and non-obese subjects at all determinations except the visit performed after 36 weeks of treatment (94.3% non-obese vs. 83.1% obese, *p* = 0.005). PASI 90 was achieved more frequently among non-obese patients than obese patients after 36 weeks (80.7% vs. 64.4%, *p* = 0.008) and 52 weeks of treatment (84.1% vs. 71.7%, 0.027). The probability of achieving PASI 75 and PASI 100 was independent of nutritional status at any time during the study. **Conclusions**: In conclusion, our results confirm that brodalumab has both rapid and sustained effectiveness in patients with moderate-to-severe psoriasis; our results could be extended to patients with multiple risk factors impairing treatment response, such as multiple biological failure and obesity.

## 1. Introduction

Plaque psoriasis is a prevalent, chronic inflammatory disease, characterized by cutaneous scaly erythematous plaques and patches typically located on the extensor surfaces, such as elbows, knees, and the lumbosacral area [1]. Plaque psoriasis carries a great disease burden due to discomfort, itching, and disfigurement, with consequent quality of life impairment and emotional issues. Additionally, it is associated with several systemic comorbidities, including psoriatic arthritis, cardiovascular diseases, diabetes, depression, and inflammatory bowel disease. Indeed, evidence from experimental and clinical studies has shown that psoriasis is a systemic condition linked to chronic inflammation [2]. Interestingly, the prevalence and severity of psoriasis appears to be associated with increased BMI [3,4]. In line with this correlation, obesity should be taken into consideration when therapies are prescribed. For example, since obese patients need higher doses of antipsoriatic drugs in comparison to non-obese patients, they suffer more complications due to adverse effects [5,6]. The goal of treatment is to manage skin lesions and enhance quality of life. While mild cases are typically controlled with topical treatments, moderate-to-severe cases require systemic therapy. Importantly, treatments should be considered based on the patient characteristics, such as comorbidities and body weight [7]. Moreover, multi-failure patients, i.e., patients who have failed multiple biologic drugs [8], have been considered. The choice of systemic treatments depends on the severity, previous therapy, comorbidities, the involvement of difficult to treat areas, and the patient’s medical history. Among the available systemic agents, the new interleukin (IL) inhibitors are highly effective and generally have a better safety profile than traditional systemic therapies [9,10]. A review of phase III clinical trials for three of the IL-17 inhibitors (secukinumab, ixekizumab, and brodalumab) highlighted that IL-17 inhibitors are effective therapeutic options for psoriasis patients [11]. Interestingly, it has recently been reported in an observation retrospective cohort study that IL-17 inhibitors appear to be more effective in decreasing systemic inflammatory markers in moderate-to-severe psoriasis in comparison to IL-23 inhibitors [12,13]. Brodalumab is a fully human monoclonal antibody against the anti-IL-17 receptor A, a key player of the th-17 axis widely involved in the psoriasis pathogenesis [2]. Brodalumab is approved in the EU for treating moderate-to-severe psoriasis in patients who are candidates for systemic therapy (SmPC Kyntheum) and in the USA for inadequate responders to other systemic therapies (SmPC Siliq). Phase II and phase III placebo-controlled clinical trials (AMAGINE-1/-2/-3) demonstrated its efficacy in patients with plaque psoriasis for up to 1 year, with superiority over ustekinumab in a head-to-head comparison and a favorable safety profile [14,15,16,17]. A post-hoc pooled analysis of AMAGINE-2 and AMAGINE-3 showed that the efficacy of brodalumab was maintained up to week 120 [18,19]. Real-life studies in patients with moderate-to-severe psoriasis, albeit with varying designs and endpoints, have shown brodalumab to produce a rapid response and long-term effectiveness up to 3 years, with a high probability of prolonged drug survival and improved quality of life [15,16,20,21,22,23]. A recent report of 5-year pharmacovigilance data confirmed the safety profile of brodalumab as demonstrated by clinical trials and previous reports [24]. Our study aimed to confirm the effectiveness and safety of brodalumab in a large real-world cohort including patients with previous biologic failures and severe body weight abnormalities.

## 2. Materials and Methods

A retrospective observational study was performed between May 2023 and June 2024 in 41 dermatological centers in Italy. Patients with moderate-to-severe plaque psoriasis treated with brodalumab were enrolled. Psoriasis was considered moderate-to-severe based on the presence of a Psoriasis Area Severity Index (PASI) ≥ 10 or less if the involvement of difficult-to-treat areas (face, palms/soles, genitalia, nails) was present. Brodalumab was administered in accordance with the Italian Guidelines for managing plaque psoriasis [25] and the summary of product characteristics in adult patients who were candidates for systemic treatments [26]. The recommended dose is 210 mg administered by subcutaneous injection at weeks 0, 1, and 2 followed by 210 mg every 2 weeks. Importantly, all patients received brodalumab as monotherapy and no concomitant systemic or topical therapies were used during the study. Institutional review board approval was waived for this study, as the procedures adopted did not deviate from standard clinical practice. All patients provided written informed consent to have their data collected during routine clinical practice, i.e., demographics and clinical parameters, included in this retrospective study, for the retrospective retrieval of anonymized data. The study was conducted in accordance with the 1964 Helsinki Declaration and its later amendments. Data collection and management observed the applicable rules, regulations, and directives concerning patient protection, such as patient privacy.

### 2.1. Data Collection

Clinical and demographic data collected at baseline included age, sex, disease duration, body mass index (BMI), comorbidities, previous exposure to biologic drugs, and the involvement of difficult-to-treat areas (scalp/face, palms/soles, genitalia, and nails). Baseline patients’ nutritional status was assessed and classified according to WHO recommendations. At baseline and at each scheduled time point (4, 16, 36, and 52 weeks), the following clinical parameters were measured: PASI, PASI improvements of 75%, 90%, and 100%, and Dermatology Life Quality Index (DLQI). For patients who missed scheduled dermatological visits or skipped the dose, the last-observation-carried-forward method was applied. At each visit, the occurrence of any adverse events (AEs; Appendix A) was recorded, including serious AEs and those leading to brodalumab discontinuation.

### 2.2. Statistical Analysis

Data were summarized by means of descriptive analysis. Means and standard deviations (SDs) were calculated for continuous variables, while absolute values and frequency (%) were calculated for categorical variables. A *t*-test or analysis of variance was performed to compare mean values, while a paired *t*-test compared mean values at different time points. The chi-squared test was used to compare frequencies. The statistical significance was defined as *p* < 0.05. All analyses were performed with IBM SPSS Statistics for Windows, Version 26.0.

## 3. Results

### 3.1. Patients’ Characteristics

The baseline demographic and clinical characteristics of patients are summarized in Table 1. Overall, 299 patients were included, 198 (66.2%) of whom were males; 63 (21.4%) were obese (BMI > 30) and 124 (42.2%) were overweight (BMI between 25 and 30).

Forty-four patients (14.7%) had joint involvement classified as psoriatic arthritis. The mean BMI was 26.6 ± 5.5 kg/m^2^. The mean duration of psoriasis was 18.1 ± 12.5 years, the mean PASI value was 15.9 ± 7.9 at baseline, and 267 (89.6%) patients exhibited difficult-to-treat areas involved.

Overall, 147 (49.2%) patients were naïve to biologic agents, and 52 (17.4%) were naïve to any systemic treatments for psoriasis. Among the 152 patients who previously received biologic therapies, 46 (30.3%) were treated with more than one agent. The last biologic preferentially used before brodalumab was adalimumab (n = 48, 31.6%) (Table 1). There were no relevant differences in demographic characteristics between patients naïve to biologics and bio-experienced patients. The baseline PASI score was 17.3 ± 8.2 in bio-naïve patients and 14.7 ± 7.4 in bio-experienced patients (*p* = 0.004).

### 3.2. Effectiveness

Overall population

The PASI score improved significantly when compared with the baseline in the study population by week 4 and continued to decrease during the study follow-up (15.9 ± 7.9 at baseline, 5.4 ± 5.3 at week 4, 1.9 ± 3.6 at week 6, 1.0 ± 2.1 at week 36, and 0.8 ± 2.1 at week 52; *p* < 0.001). PASI 75 was achieved by 131/299 (43.8%) patients at week 4, 233/297 (78.5%) patients at week 16, and 268/292 (91.8%) at week 36; this proportion was maintained at 52 weeks. PASI 90 was reached by 60/299 (20.1%) patients at week 4, 194/297 (65.3%) at week 16, 225/292 (77.1%) at week 36, and 238/292 (81.5%) at week 52. A complete skin clearance, defined as PASI 100, was observed in 45/299 (15.1%) patients at week 4, 163/297 (54.9%) patients at week 16, 183/292 (62.7%) at week 36, and 204/292 (69.9%) at week 52 (Table 2). The PASI score was ≤2 in 99/297 (33.1%) at week 4, 214/292 (72.1%) at week 16, 257/292 (88.0%) at week 36, and 263/299 (90.1%) at week 52.

The speed of action and the consistent effectiveness of brodalumab on the skin were accompanied by a clear improvement in patients’ quality of life. The mean DLQI score decreased at 4.8 ± 5 at week 4, 2.0 ± 3.4 at week 16, 1.3 ± 3.3 at week 36 and 1.0 ± 3.0 at week 52 with only one patient scoring > 21 after 1 year of treatment.

Impact of previous biologics

In both bio-naïve and bio-experienced patients, a significant improvement in PASI score was observed throughout the study. The mean PASI for bio-naïve patients decreased from 17.3 ± 8.2 at baseline to 1.0 ± 2.5 at 52 weeks, with a significant reduction noted at 4 weeks (5.6 ± 5.5), 16 weeks (2.0 ± 3.6), and 36 weeks (1.1 ± 2.4) (*p* < 0.001). Bio-experienced patients’ PASI improved to 0.7 ± 1.6 at 52 weeks, with mean PASI scores of 5.2 ± 5.1 at 4 weeks, 1.9 ± 3.6 at 16 weeks, and 0.9 ± 1.8 at 36 weeks (*p* < 0.001 at all time points). Although baseline PASI was significantly higher in naïve patients vs. experienced patients, the same difference was not observed during the follow-up time points. The trend in PASI 75, 90, and 100 achievement was comparable among bio-naïve and bio-experienced patients across all time-points. The percentage of patients reaching PASI ≤ 2 was high and similar among both groups at each scheduled visit (*p*-values > 0.46).

Impact of body weight

Mean PASI score did not differ significantly in obese patients (16.6 ± 9.0, n = 63) as compared with normal-weight and overweight non-obese patients (15.9 ± 7.6, n = 231) at baseline, nor after 4 and 16 weeks, but was significantly higher in obese patients after 36 weeks (1.7 ± 2.5 vs. 0.8 ± 2.0, *p* = 0.009) and after 52 weeks (1.4 ± 2.6 vs. 0.7 ± 1.9, *p* = 0.026). Nevertheless, PASI score reduction was significant at each follow-up visit for both obese and non-obese patients (*p* < 0.001 in both groups at each time point). Non-obese patients had a higher likelihood of PASI 75 achievement only at week 36 (94.3% of non-obese vs. 83.1% obese, *p* = 0.005), and of PASI 90 achievement at week 36 (80.7% vs. 64.4%, *p* = 0.008) and week 52 (84.1% vs. 71.7%, *p* = 0.027), while the probability of obtaining PASI 100 was similar at any time during the study follow-up (Table 3, Figure 1).

### 3.3. Safety

Overall, 15 (5.0%) patients experienced a total of 17 AEs during the observation period of treatment, including positivity of antinuclear antibodies, candidiasis of the oral cavity, non-specific colitis, lower limb edema, hypotension, maculopapular rash, epilepsy, non-specific diffuse erythema, fatigue, oral leucoplakia, and steatosis, each one occurring in one patient (5.9%), with the most frequent being the detection of latent tuberculosis (4/17; 23.5%) and the onset of conjunctivitis (2/17; 11.8%) (Appendix A). Therapy was discontinued by 15 patients (5.0%) for various reasons [FIVET and pregnancy research (n = 3, 1.0%); loss of efficacy (n = 3, 1.0%); PSA worsening (n = 1, 0.3%); loss of joint efficacy (n = 1, 0.3%); patient lost at follow-up (n = 1, 0.3%); unknown (n = 1, 0.3%)], including AEs (edema of the lower limbs, hypotension, and maculopapular rash; fatigue; non-specific colitis; latent tuberculosis; leucoplakia) in 5/15 (1.7%) patients. Information on the timing of discontinuation was not available.

## 4. Discussion

This retrospective study in a real-life setting confirmed the rapid and sustained effectiveness of brodalumab up to 1 year of treatment in patients with moderate-to-severe psoriasis. A significant reduction in mean PASI score was obtained at 4 weeks, regardless of previous treatments with other biological agents and the presence of obesity; the improvement achieved was maintained in the overall cohort and in subgroups for up to 1 year. An improvement in quality of life was also observed. Both obese and non-obese patients obtained significant reductions of PASI and a similar probability of achieving PASI 100 during the follow-up. On the contrary, starting from week 36, the potential of obtaining a significant PASI improvement was influenced by obesity. Our results suggest that obese subjects and patients who have already failed treatment with a biologic agent may benefit from treatment with brodalumab as well as non-obese and bio-naïve ones. The efficacy data observed in this study are in agreement with data from clinical trials [27]. Real-world data are obtained in non-selected populations, including subjects with comorbidities and older age, and are liable to greater variability than clinical trial evidence. On the other hand, they better reflect clinical practice and provide information for clinical decisions in those conditions excluded from trials. Comparison with real-world studies is not direct, because different study designs are used, and the absence of a strict selection may introduce a number of biases. Nevertheless, data from real-world studies will help to optimize and customize the use of available drugs. The efficacy outcomes, in terms of probability of PASI 75, 90, and 100 achievement at 1 year, observed in our patients are similar to those obtained in a Polish register study (92.5% vs. 95.1%, 81.5% vs. 87.3%, and 69.9% vs. 69.7%, respectively) [20]. In both studies, PASI improvement was associated with an improvement in quality of life (DLQI score). Kojanova et al. also found that drug survival probability at 1 year was 84.2%, and that longer drug survival was associated with younger age, higher BMI, and no previous biologic treatment. Indeed, Caldarola et al. [28] found in an Italian retrospective study that higher BMI was predictive for treatment discontinuation, suggesting some difference between this population and the one observed by Kojanova et al. In a German study, the frequency of PASI 75, 90, and 100 increased from 3 months to 1 year of treatment (PASI 100, 37% at 3 months and 51.5% at 1 year), in agreement with our observations [PASI 100, 54.9 at 4 months and 69.9% at 1 year] [21]. A Greek group analyzed data based on the “as observed data”, reporting that 42% of patients obtained PASI 100 after 25.9 ± 3.5 weeks, and 65% of patients achieved PASI 100 at Week 104 [22]. This type of analysis can be particularly useful for supporting physicians in guiding patients’ expectations of treatment effects. Obesity in psoriasis patients is associated with increased skin lesion severity and reduced response to therapy [29]. These events are attributed to the low-level chronic inflammation observed in obese patients and characterized by the higher production of pro-inflammatory cytokines IL-23 and IL-17 [30,31]. Brodalumab, unlike other biologics, does not require dose adjustment based on body weight [32,33]. In a post-hoc analysis of AMAGINE-2 and AMAGINE-3, including patients with obesity (BMI ≥ 30 kg/m^2^) and without obesity (BMI < 30 kg/m^2^), the rates of PASI 75, PASI 90, and PASI 100 achieved with continuous treatment with brodalumab at week 12 were independent of obesity status (>97%, >74%, and >49%, respectively), and the differences between the groups were not significant at week 52 [33]. Our real-life observations confirm obesity does not affect brodalumab performance in psoriasis patients. Previous studies suggested that the efficacy of biologics may be reduced after the failure of previous biologics, although data are not always consistent [34,35]. In our experience, we observed that bio-experienced patients had similar improvement from brodalumab treatment compared to bio-naive ones. Limitations of this study include the observational design (which is inherent to a real-life study, and open to special considerations), the follow-up limited to one year, the retrospective design, and the absence of data on drug survival. Nevertheless, other studies have shown the long-term drug survival rate, e.g., a recent study by Mastorino and colleagues that reported the safety profile of brodalumab in a real-world setting for up to 3 years of treatment [36]. Concerning the sample size, it is necessary to underline that due to the retrospective and real-world design of this study, the sample was determined by the data available during the study period. Therefore, this determined the a priori sample size calculation. In addition, the study focused on ensuring the real-world population’s representativeness rather than using a prospective design. Therefore, despite the methodological limitation of retrospective studies, we believe that the findings are clinically meaningful and provide valuable insights applicable to daily practice. Indeed, these findings might be considered in clinical practice in the choice of the best treatment in case of patients for whom the standard therapies are not suitable, e.g., obese and multi-failure patients [8,27].

## 5. Conclusions

This observational study in a real-life setting confirms that brodalumab has a rapid onset of action and prolonged effectiveness in patients with moderate-to-severe psoriasis; these results may also be extended to subjects with risk factors for reduced response to biologic treatment, such as previous biologic failures and obesity.

## Figures and Tables

**Figure 1 jcm-14-01087-f001:**
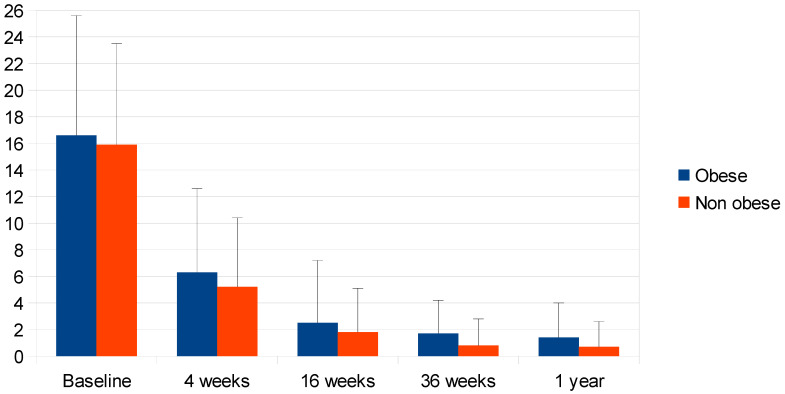
Mean PASI of obese and non-obese patients from baseline to 1 year from the treatment, as shown in Table 3 (Mean ± SEM).

**Table 1 jcm-14-01087-t001:** Clinical and demographic characteristics at baseline.

Characteristics	n = 299	Bio-Naï ven = 147	Bio-Experienced n = 152	*p* Value	Obese n = 63	Non-Obese n = 231	*p* Value
Male, n (%) Female	198 (66.2%) 101 (33.8%)	102 (69.4%) 45 (30.6%)	96 (63.2%) 56 (36.8%)	0.256	43 (68.3%) 20 (31.7%)	153 (66.2%) 78 (33.8%)	0.763
Age (years), mean ± SD	53.0 (±14.5)	51.5 (±15.0)	54.4 (±13.9)	0.092	53.5 (±14.4)	52.9 (±14.6)	0.770
Disease duration (years), mean ± SD	18.1 (±12.5)	16.8 (±12.3)	19.3 (±12.5)	0.050	18.0 (±12.0)	18.1 (±12.7)	0.963
BMI, mean ± SD	26.6 (±5.5)	27.5 (±5.8)	26.7 (±5.2)	0.159	35.2 (±4.6)	24.9 (±3.1)	<0.001
Obese, n (%)	63 (21.4%)	32 (21.8%)	31 (20.4%)	0.700	63 (100%)	-	<0.001
Overweight, n (%)	124 (42.2%)	63 (42.9%)	61 (40.1%)	0.526	-	124 (53.7%)	<0.001
Normal weight, n (%)	99 (33.7%)	42 (28.6%)	57 (37.5%)	0.129	-	99 (42.9%)	<0.001
Comorbidities							
• Diabetes, n (%) • Hypertension, n (%) • Hyperlipidemia, n (%) • Thyroid disease, n (%) • Oncologic disease, n (%) • Cardiopathy, n (%) • Other, n (%)	36 (12.0%) 119 (39.8%) 92 (30.8%) 18 (6.0%) 10 (3.3%) 31 (10.4%) 81 (27.1%)	22 (15.0%) 48 (32.7%) 45 (30.6%) 7 (4.8%) 6 (4.1%) 19 (12.9%) 41 (27.9%)	14 (9.2%) 71 (46.7%) 47 (30.9%) 11 (7.2%) 4 (2.6%) 12 (7.9%) 40 (26.3%)	0.127 0.013 0.954 0.369 0.486 0.154 0.760	16 (25.4%) 36 (57.1%) 27 (42.9%) 3 (4.8%) - 7 (11.1%) 28 (44.4%)	20 (8.7%) 82 (35.5%) 65 (28.1%) 14 (6.1%) 10 (4.3%) 24 (10.4%) 51 (22.1%)	<0.001 0.002 0.026 0.696 0.094 0.869 <0.001
PsA, n (%)	44 (14.7%)	22 (15.0%)	22 (14.5%)	0.905	10 (15.9%)	33 (14.3%)	0.752
≥1 Difficult-to-treat areas, n (%)	267 (89.6%)	136 (92.5%)	131 (86.8%)	0.104	52 (82.5%)	212 (92.2%)	0.024
PASI baseline, mean ± SD	15.9 (±7.9)	17.3 (±8.2)	14.7 (±7.4)	0.004	16.6 (±9.0)	15.9 (±7.6)	0.519
Bio-naïve, n (%)	147 (49.2%)	147 (100%)	-	<0.001	32 (50.8%)	111 (48.1%)	0.700
Naïve for systemic therapies, n (%)	52 (17.4%)	41 (27.9%)	11 (7.2%)	<0.001	14 (22.2%)	33 (14.3%)	0.128
Last biologic used before brodalumab, n (%)							
• Adalimumab • Etanercept • Secukinumab • Apremilast • Ustekinumab • Ixekinumab • Guselkumab • Tildrakizumab • Risankizumab • Infliximab • Bimekizumab	48 (31.6%) 31 (20.4%) 24 (15.8%) 11 (7.2%) 9 (5.9%) 9 (5.9%) 6 (3.9%) 5 (3.3%) 4 (2.6%) 3 (2.0%) 1 (0.7%)		48 (31.6%) 31 (20.4%) 24 (15.8%) 11 (7.2%) 9 (5.9%) 9 (5.9%) 6 (3.9%) 5 (3.3%) 4 (2.6%) 3 (2.0%) 1 (0.7%)		10 (32.3%) 4 (12.9%) 3 (9.7%) 3 (9.7%) 2 (6.5%) 3 (9.7%) 2 (6.5%) 2 (6.5%) 1 (3.2%) 1 (3.2%) -	38 (31.7%) 27 (22.5%) 21 (17.5%) 8 (6.7%) 7 (5.8%) 6 (5.0%) 4 (3.3%) 3 (2.5%) 3 (2.5%) 2 (1.7%) -	

BMI (Body Mass Index); PsA (Psoriatic Arthritis); PASI (Psoriasis Areas Severity Index).

**Table 2 jcm-14-01087-t002:** Proportion of patients achieving PASI 75, 90, and 100 during the study follow-up.

**PASI 75**		
**Visit**	**n**	**%**
4 weeks	131/299	43.8
16 weeks	233/297	78.5
36 weeks	268/292	91.8
52 weeks	270/292	92.5
**PASI 90**		
**Visit**	**n**	**%**
4 weeks	60/299	20.1
16 weeks	194/297	65.3
36 weeks	225/292	77.1
52 weeks	238/292	81.5
**PASI 100**		
**Visit**	**n**	**%**
4 weeks	45/299	15.1
16 weeks	163/297	54.9
36 weeks	183/292	62.7
52 weeks	204/292	69.9
**PASI ≤2**		
**Visit**	**n**	**%**
Baseline	2/299	0.7
4 weeks	99/297	33.1
16 weeks	214/292	72.1
36 weeks	257/292	88.0
52 weeks	263/299	90.1

**Table 3 jcm-14-01087-t003:** Mean PASI score in bio-naïve and bio-experienced, obese and non-obese patients from baseline to 1 year from the treatment.

PASI	Bio-Naï ven = 147	Bio-Experienced n = 152	Independent Samples *t*-test
Baseline	17.3 (±8.2)	14.7 (±7.4)	0.004
4 weeks	5.6 (±5.5)	5.2 (±5.1)	0.476
16 weeks	2.0 (±3.6)	1.9 (±3.6)	0.856
36 years	1.1 (±2.4)	0.9 (±1.8)	0.253
1 year	1.0 (±2.5)	0.7 (±1.6)	0.285
	**Obese** **n = 63**	**Non Obese** **n = 231**	**Independent Samples *t*-test**
Baseline	16.6 (±9.0)	15.9 (±7.6)	0.519
4 weeks	6.3 (±6.3)	5.2 (±5.0)	0.133
16 weeks	2.5 (±4.7)	1.8 (±3.3)	0.158
36 years	1.7 (±2.5)	0.8 (±2.0)	0.009
1 year	1.4 (±2.6)	0.7 (±1.9)	0.026

## Data Availability

Additional data supporting the findings of this study are available from the Corresponding Author on reasonable request.

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
