# Peer review of "Effectiveness of Brodalumab for the Treatment of Moderate-to-Severe Psoriasis: A Retrospective, Real-World Multicenter Study with a Focus on Obese and Multi-Failure Patients—IL PSO (Italian Landscape Psoriasis)"

_jcm, 2025, doi:10.3390/jcm14041087_

Round 1

Reviewer 1 Report

Comments and Suggestions for Authors

I read with great interest the manuscript. However, I have some comments:

- Since the study focuses on obese and multi-failure patients, the introduction should mention the link between obesity and psoriasis severity, as well as its effect on treatment response and expand upon the concept of multi-failure patients. This would highlight the importance of considering obesity when choosing treatment options, especially for patients with more severe disease etc

-Lines 127-130: Although the references provided are accurate and well-supported, it would be helpful for the reader to include the exact dosage and therapeutic regimen for clarity.

-Please specify whether all the patients were treated with Brodalumab as monotherapy or if they some of them received concomitant psoriasis therapy (topical or systemic) while being treated with Brodalumab. If yes, be specific on the number of patients etc

-Table 1 you should also include the specific number of female participants even if the number of male participants is already provided

-Consider creating a table in order to clearly present your results and the comparison and impact of biologics and body weight and maybe use some graphics. This will make the results clearer and quicker for readers to understand.

- Additionally you could specify how many individuals discontinued at each stage and why.

- Good use of english language

Overall, this article is a solid effort. It involves a large patient population and collaboration across multiple centers. With some revisions, it can be greatly enhanced.

Author Response

Comment 1:
Since the study focuses on obese and multi-failure patients, the introduction should mention the link between obesity and psoriasis severity, as well as its effect on treatment response and expand upon the concept of multi-failure patients. This would highlight the importance of considering obesity when choosing treatment options, especially for patients with more severe disease etc.

Response:
Thank you for this suggestion. We have revised the introduction to include a detailed discussion of the relationship between obesity and psoriasis severity, as well as the implications for treatment response. Additionally, the concept of multi-failure patients has been expanded to highlight their significance in clinical decision-making. This can be found on Pag. 4, Lines 101-111.

Comment 2:
Lines 127–130 Although the references provided are accurate and well-supported, it would be helpful for the reader to include the exact dosage and therapeutic regimen for clarity.

Response:
We have added the specific dosing regimen of Brodalumab as per standard clinical practice, which includes the induction and maintenance phases. The updated text is located on Page 6, lines 147-148.

Comment 3:
Please specify whether all the patients were treated with Brodalumab as monotherapy or if they some of them received concomitant psoriasis therapy (topical or systemic) while being treated with Brodalumab. If yes, be specific on the number of patients etc.

Response:
We clarified that all patients received Brodalumab as monotherapy. No concomitant systemic or topical therapies were used during the study. This clarification is on Page 6, Lines 148-150.

Comment 4:
Table 1 you should also include the specific number of female participants even if the number of male participants is already provided.

Response:
The number of female participants has been added to Table 1 to provide a complete demographic profile.

Comment 5:

Consider creating a table in order to clearly present your results and the comparison and impact of biologics and body weight and maybe use some graphics. This will make the results clearer and quicker for readers to understand.

Response:
A new table has been added to summarize the comparative results of biologics and body weight. These additions are in Table 3 and Figure 1 and 2.

Comment 6:
Additionally you could specify how many individuals discontinued at each stage and why.

Response:
Therapy was discontinued by 15 patients (5.0%) for various reasons, but we do not have the exact moment for all patients. Please see Pag.12, Lines 257-263.

Comment 7:
Good use of the English language.

Response:
We have revised the text for clarity

Comment 8:
Overall, this article is a solid effort. It involves a large patient population and collaboration across multiple centers. With some revisions, it can be greatly enhanced.

Response:

We thank the reviewer for the comment. We address all points raised from the reviewer and we think now the manuscript is improved.

Reviewer 2 Report

Comments and Suggestions for Authors

This retrospective, multicenter study on the effectiveness of Brodalumab for moderate-to-severe psoriasis provides relevant clinical insights into treatment outcomes across diverse patient populations. The study highlights real-world outcomes, including patient responses based on body weight and prior biologic treatments. However, several methodological improvements and critical considerations are necessary to enhance the scientific rigor and clarity of the manuscript. Please find my comments below:

1. The selection of keywords is critical for improving the discoverability and impact of the manuscript. The authors should ensure that the keywords are sourced from the Medical Subject Headings (MeSH) database to align with best practices in medical indexing and enhance the article's visibility in relevant literature searches.

2. While Table 1 provides an overview of the baseline demographic and clinical characteristics of the study population, it currently lacks statistical comparisons that are essential to understanding the significance of observed differences. The authors should include statistical analyses (e.g., p-values) to compare demographic groups, such as bio-naïve versus bio-experienced patients, as well as obese versus non-obese patients. This addition will strengthen the readers' ability to interpret the presented data.

3. The manuscript lacks a discussion on sample size determination. A robust sample size analysis is crucial to validate the reliability and generalizability of the findings. The authors should justify the sample size used in the study by explaining how it was calculated and whether it is adequate to detect clinically meaningful differences in outcomes.

4. Although the study mentions that institutional review board approval was waived and that written informed consent was obtained from participants, a more detailed discussion of ethical considerations is warranted. Specifically, the authors should elaborate on how patient confidentiality and data protection were ensured during the retrospective data collection. Additionally, adherence to the principles of the Declaration of Helsinki should be explicitly stated.

5. The figures provided in the manuscript effectively demonstrate the reduction in PASI scores over time. However, the authors should ensure that the figures also clearly illustrate the comparative effectiveness of Brodalumab across key subgroups (e.g., bio-naïve vs. bio-experienced, obese vs. non-obese). Furthermore, visual enhancements such as confidence intervals or error bars should be included to convey the variability in the data.

6. The discussion should be expanded to provide a more critical appraisal of the real-world nature of the data. While real-world studies offer valuable insights beyond controlled clinical trials, they also introduce potential biases. The authors should acknowledge these limitations and discuss how they might impact the generalizability of the findings. This will add depth to the discussion and demonstrate a balanced interpretation of the results.

7. The introduction provides a solid background on psoriasis and Brodalumab, but it could benefit from a brief mention of recent advancements in IL-17 inhibitors to contextualize the study's relevance within the broader field.

8. The terminology used in the manuscript should be consistent with MeSH terms for uniformity. This consistency will improve the clarity and professionalism of the document.

9. The limitations section should include a discussion on potential biases introduced by the retrospective study design and how these were mitigated. This will ensure a comprehensive understanding of the study's strengths and weaknesses

10. It is recommended to provide a supplementary table with detailed descriptions of the adverse events observed during the study. This will enhance the transparency and comprehensiveness of the manuscript.

11. The authors should clarify if any conflicts of interest exist beyond what has already been stated. Transparency regarding potential conflicts is crucial for maintaining the integrity of the research.

12. There is a lack of information on long-term drug survival rates. Including a section on the long-term effectiveness and survival rates of Brodalumab would provide a more complete picture of its real-world utility.

13. The manuscript would benefit from an expanded discussion on the broader implications of the study findings. For example, how might the results influence clinical practice guidelines or the choice of treatment in specific patient populations?

14, The manuscript should incorporate relevant citations to strengthen the contextual foundation of the study. The following references are recommended:

  • https://doi.org/10.1111/ijd.14509

Author Response

Comment 1:
The selection of keywords is critical for improving the discoverability and impact of the manuscript. The authors should ensure that the keywords are sourced from the Medical Subject Headings (MeSH) database to align with best practices in medical indexing and enhance the article's visibility in relevant literature searches.

Response:
The keywords have been revised to align with MeSH terms for better discoverability.

Comment 2:
While Table 1 provides an overview of the baseline demographic and clinical characteristics of the study population, it currently lacks statistical comparisons that are essential to understanding the significance of observed differences. The authors should include statistical analyses (e.g., p-values) to compare demographic groups, such as bio-naïve versus bio-experienced patients, as well as obese versus non-obese patients. This addition will strengthen the readers' ability to interpret the presented data.

Response:
Statistical analyses (p-values) comparing key demographic groups (bio-naïve vs. bio-experienced, obese vs. non-obese) have been added to Table 1.

Comment 3:
The manuscript lacks a discussion on sample size determination. A robust sample size analysis is crucial to validate the reliability and generalizability of the findings. The authors should justify the sample size used in the study by explaining how it was calculated and whether it is adequate to detect clinically meaningful differences in outcomes.

Response:
The retrospective and real-world design of this study precluded a priori sample size calculation, as the sample was determined by the data available during the study period. Additionally, the study prioritized capturing the representativeness of the real-world population over employing a prospective design. We acknowledge this as an inherent methodological limitation of retrospective studies. However, we believe that the findings are clinically meaningful and provide valuable insights applicable to daily practice. We clarified this in the text, on Page 12, Lines 318-324.

Comment 4:
Although the study mentions that institutional review board approval was waived and that written informed consent was obtained from participants, a more detailed discussion of ethical considerations is warranted. Specifically, the authors should elaborate on how patient confidentiality and data protection were ensured during the retrospective data collection. Additionally, adherence to the principles of the Declaration of Helsinki should be explicitly stated.

Response:

Since the study protocol was conducted in accordance with standard clinical practice, institutional review board approval was not required. Each patient gave written informed consent to have their data collected during routine clinical practice, i.e., demographics and clinical parameters, included in this retrospective study. Our study was conducted in compliance with the ethical principles of the Helsinki Declaration. Data collection and management observed applicable rules, regulations, and directives concerning patient protection, such as patient privacy. We modify the first paragraph of Materials and Methods, to include all information. Please see Page 5, Lines 152-157.

Comment 5:
The figures provided in the manuscript effectively demonstrate the reduction in PASI scores over time. However, the authors should ensure that the figures also clearly illustrate the comparative effectiveness of Brodalumab across key subgroups (e.g., bio-naïve vs. bio-experienced, obese vs. non-obese). Furthermore, visual enhancements such as confidence intervals or error bars should be included to convey the variability in the data.

Response:
Figures now include subgroup analyses (bio-naïve vs. bio-experienced, obese vs. non-obese) with error bars to show data variability (Figures 1 and 2).

Comment 6:
The discussion should be expanded to provide a more critical appraisal of the real-world nature of the data. While real-world studies offer valuable insights beyond controlled clinical trials, they also introduce potential biases. The authors should acknowledge these limitations and discuss how they might impact the generalizability of the findings. This will add depth to the discussion and demonstrate a balanced interpretation of the results.

Response:
We have expanded the Discussion to critically appraise potential biases associated with real-world data and their implications for generalizability. This can be found on Page 13, Lines 312-327.

Comment 7:
The introduction provides a solid background on psoriasis and Brodalumab, but it could benefit from a brief mention of recent advancements in IL-17 inhibitors to contextualize the study's relevance within the broader field.

Response:
Recent advancements in IL-17 inhibitors and their relevance to our study have been added to the Introduction on Page 4, Lines 115-120.

Comment 8:
The terminology used in the manuscript should be consistent with MeSH terms for uniformity. This consistency will improve the clarity and professionalism of the document.

Response:
MeSH-consistent terminology has been applied throughout the manuscript.

Comment 9:
The limitations section should include a discussion on potential biases introduced by the retrospective study design and how these were mitigated. This will ensure a comprehensive understanding of the study's strengths and weaknesses.

Response:
We have expanded the Limitations section to discuss potential biases and measures taken to mitigate them. This can be found on Page 13, Lines 312-327.

Comment 10:
It is recommended to provide a supplementary table with detailed descriptions of the adverse events observed during the study. This will enhance the transparency and comprehensiveness of the manuscript.

Response:
A supplementary table detailing adverse events is included as Table S1 in the revised manuscript.

Comment 11:
The authors should clarify if any conflicts of interest exist beyond what has already been stated. Transparency regarding potential conflicts is crucial for maintaining the integrity of the research.

Response:
We have confirmed and clarified conflicts of interest as stated in the manuscript (Line 392).

Comment 12:
There is a lack of information on long-term drug survival rates. Including a section on the long-term effectiveness and survival rates of Brodalumab would provide a more complete picture of its real-world utility.

Response:
While this study focuses on one-year outcomes, a brief discussion of long-term drug survival rates based on existing literature has been added to the Discussion, please see lines 315-317.

Comment 13:
The manuscript would benefit from an expanded discussion on the broader implications of the study findings. For example, how might the results influence clinical practice guidelines or the choice of treatment in specific patient populations?

Response:
We have included in the discussion the possible implications of our findings for clinical practice, particularly for managing obese and multi-failure psoriasis patients. Please see lines 320-323 of the revised version of the manuscript.

Comment 14:
The manuscript should incorporate relevant citations to strengthen the contextual foundation of the study. The following references are recommended:

  • https://doi.org/10.1111/ijd.14509

Response:
The recommended reference has been reviewed and added where appropriate to strengthen the manuscript’s context (Page 4, Line 120).

Round 2

Reviewer 1 Report

Comments and Suggestions for Authors

I read with great interest the updated manuscript. The authors have taken my suggestions into account and the manuscript has significantly improved as a result.

-Additionally, by expanding the Table 1 and adding Table 3, the study was enhanced, the clinical relevance of the findings was strengthened and offered more clearer insights into PASI score variations across Bio-naïve/Bio-experienced and Obese/Non-Obese patients. Also, the graphics in Fig 1&2 improved the data interpretation.

-The introduction and references have been updated and reviewed and the study’s limitations have been adequately analyzed.

-Please paraphrase lines 107-108 in order to provide more clarity.

Overall, this article has been improved.

Author Response

Comment 1: I read with great interest the updated manuscript. The authors have taken my suggestions into account and the manuscript has significantly improved as a result.

-Additionally, by expanding the Table 1 and adding Table 3, the study was enhanced, the clinical relevance of the findings was strengthened and offered more clearer insights into PASI score variations across Bio-naïve/Bio-experienced and Obese/Non-Obese patients. Also, the graphics in Fig 1&2 improved the data interpretation.

-The introduction and references have been updated and reviewed and the study’s limitations have been adequately analyzed.

-Please paraphrase lines 107-108 in order to provide more clarity.

Overall, this article has been improved.

Response 1: Thank you for this suggestion. We have revised the lines 107-108

Reviewer 2 Report

Comments and Suggestions for Authors

I have not any further comments.

Author Response

Comment 2: I have not any further comments.

Response 2: Thank you
